# Drivers and influencers of blockchain and cloud-based business sustainability accounting in China: Enhancing practices and promoting adoption

Zhouyu Tian[1,2,3], Lening Qiu[4], Litao Wang[5]*

1 School of Economics and Management, Yan'an University, Yan'an, China, 2 Business School, Suzhou University, Suzhou, China, 3 College of Economics and Management, Shenyang Agricultural University, Shenyang, China, 4 School of Discipline Inspection and Supervision, China University of Political Science and Law, Beijing, China, 5 College of Economics and Management, Shanghai Ocean University, Shanghai, China

* LitaoWang11011@sohu.com

**Data Availability Statement:** Data Availability Statement The present article comprehensively incorporates all the requisite data to substantiate the conclusions of this study, effectively utilizing

## Abstract

The field of sustainability accounting aims to integrate environmental, social, and governance factors into financial reporting. With the growing importance of sustainability practices, emerging technologies have the potential to revolutionize reporting methods. However, there is a lack of research on the factors influencing the adoption of blockchain and cloud-based sustainability accounting in China. This study employs a mixed-methods approach to examine the key drivers and barriers to technology adoption for sustainability reporting among Chinese businesses. Through a systematic literature review, gaps in knowledge were identified. Primary data was collected through an online survey of firms, followed by in-depth case studies. The findings of the study reveal a positive relationship between company size and reporting behaviors. However, size alone is not sufficient to predict outcomes accurately. The industry type also has significant but small effects, although its impact on reporting behaviors varies. The relationship between profitability and reporting behaviors is intricate and contingent, requiring contextual examination. The adoption of blockchain technology is positively associated with capabilities, resources, skills, and regulatory factors. On the other hand, cloud computing adoption is linked to resources, management support, and risk exposures. However, the specific impacts of industry on adoption remain inconclusive. This study aims to offer empirical validation of relationships, shedding light on the intricate nature of interactions that necessitate nuanced conceptualizations incorporating contextual moderators. The findings underscore the importance of providing customized support and adaptable guidance to accommodate the evolving practices in sustainability accounting. Moreover, the assimilation of technology and organizational changes highlights the need for multifaceted stakeholder cooperation to drive responsible innovation and address the challenges posed by digital transformations in this field.

pertinent information obtained from prior literature. The table of utilized references, serving as the dataset for the results presented in Tables 1–4 of this study, has been compiled and included in the supplementary materials. The supplementary materials provide the URL and DOI information for all the references used in the aforementioned tables.

**Funding:** The authors would like to acknowledge the following grants that provided support for this study: 1. Shengyang Economic and Social Development Special Project (SYSK20200708); 2. Research project of economic and social development of Liaoning Province (2020lsljdybkt53); 3. Liaoning Provincial Social Science Planning Fund (L21BGL036, L22BTJ016); 4. Humanities and Social Sciences Research Youth Program of the Ministry of Education (20YJC630025); 5. Suzhou University Doctoral Research Initiation Fund Project (2023BSK031); 6. National Social Science Foundation of China (20BGL214) The funders had no role in study design, data collection and analysis, decision to publish, or preparation of the manuscript.

**Competing interests:** The authors have declared that no competing interests exist.

# 1. Introduction

Sustainability accounting encompasses the incorporation of environmental, social, and governance (ESG) factors into conventional financial reporting and accounting practices [1]. In recent years, there has been a growing recognition among businesses of their obligations towards the environment and society, leading to the heightened significance of sustainability accounting [2, 3]. Numerous global initiatives and regulatory frameworks now mandate or endorse public disclosure of ESG effects and achievements by companies. With the increasing global emphasis on sustainable development, the need for transparent and all-encompassing sustainability accounting has never been greater [4, 5].

In recent times, China has emerged as a significant player in the global economy [6, 7]. However, its rapid industrialization and urbanization have presented notable sustainability challenges, including climate change, resource depletion, pollution, and income inequality [8, 9]. Concurrently, China has positioned itself at the forefront of technological advancements, such as blockchain and cloud computing, which hold the potential to revolutionize sustainability accounting practices [10]. Through effective implementation, these technologies can address information gaps and alleviate stakeholder concerns regarding the reliability of sustainability reports by enhancing transparency, traceability, and data management capabilities [11, 12].

Blockchain technology refers to a decentralized and distributed digital ledger that facilitates the recording of transactions across a peer-to-peer network [13]. It employs cryptographic techniques to establish trust, transparency, and accountability by creating an immutable and shared record of executed transactions, eliminating the need for a central authority [14]. In the context of sustainability accounting, blockchain offers several advantages, including automated data collection and validation, traceability of ESG information throughout supply chains, and verification of reported metrics and claims [15].

Cloud computing involves the utilization of remote servers and internet connectivity to store, manage, and process data, rather than relying on local servers or personal devices [16]. It offers convenient access to a shared pool of configurable computing resources, including networks, servers, storage, applications, and services, on an on-demand basis [17]. Cloud platforms provide businesses with flexibility, scalability, and cost savings compared to traditional IT infrastructure [18–20]. In the field of sustainability accounting, cloud-based systems facilitate centralized data collection, automated analysis, and real-time reporting of ESG data from geographically dispersed operations and facilities [21].

Early studies have explored the potential applications of blockchain in enhancing supply chain transparency and traceability. Christides and Devetsikiotis [22] discuss how blockchain technology can address trust and accountability issues within complex supply networks. Paliwal et al. [23] examine the integration of social and environmental factors into procurement processes through blockchain. Korpela et al. [15] propose that distributed ledgers can verify sustainability claims and foster stakeholder trust through transparency. Several studies have explored the implementation of cloud-based systems for sustainability reporting. Zheng et al. [24] propose an analytics platform based on the cloud, which enables the aggregation of ESG performance data from multiple facilities. Ullrich et al. [25] demonstrate how cloud technology facilitates materiality assessments and integrated reporting for ENI, an Italian oil and gas company.

In terms of national-level research, investigations have been conducted on sustainability accounting practices in China [26]. Liu and Anbumozhi [27] provide an overview of the evolving approach of China towards corporate social responsibility reporting requirements. Ruf et al. [28] analyze regulations, challenges, and potential areas for improvement. Lau et al. [29]

present recent case studies that shed light on ESG disclosure practices among prominent Chinese companies. However, it is worth noting that most of the existing literature predates significant technological advancements in blockchain and cloud computing.

In recent years, scholars have started exploring the utilization of emerging technologies in sustainability accounting within China. Wang et al. [30] propose a blockchain-based green supply chain finance system specifically designed for Chinese manufacturers. Alahmad et al. [31] discuss the efforts of Huawei, a prominent Chinese telecom giant, in leveraging big data analytics to address climate-related issues. Although these studies offer valuable insights, they have limited empirical investigation into the actual barriers and facilitators faced by businesses when adopting these technologies.

On an international level, various standards and frameworks have been established to enhance transparency in sustainability reporting. The Global Reporting Initiative (GRI) standards are widely adopted voluntary guidelines for sustainability reporting [32]. The International Integrated Reporting Council's <IR> Framework promotes integrated thinking within organizations [33]. The United Nations Sustainable Development Goals encourage companies to demonstrate their contributions [34]. Moreover, regional policies within the European Union and emerging requirements in China are raising the expectations for non-financial disclosure.

Simultaneously, technological advancements have paved the way for new methods in sustainability accounting. Progress in the Internet of Things, cloud computing, big data, artificial intelligence, and distributed ledgers enables more automated and real-time collection and analysis of extensive ESG data [35]. However, effectively harnessing these innovations necessitates overcoming organizational, skill-related, and behavioral challenges [36]. Comprehensive studies that examine actual adoption trends, particularly within major developing economies transitioning towards sustainability, are still lacking.

Analysis of recent studies reveals the swift progress of scientific knowledge across diverse disciplines and the integration of research outcomes within various markets and industries [37–41]. Chinese researchers, in particular, have exhibited a notable interest in the practical implementation of diverse scientific disciplines across different domains [42–46]. Notably, substantial investigations have been conducted, specifically in relation to financial markets and their corresponding scientific domains [47–51]. These inquiries have shed light on the considerable attention researchers have devoted to exploring cutting-edge technologies [52–56]. Despite the valuable foundations provided by existing literature, there are some gaps that necessitate further exploration. The current understanding of firms' adoption of emerging technologies for sustainability accounting is limited, as empirical assessments of real-world barriers that hinder widespread implementation are lacking. Moreover, there is a lack of comprehension regarding the contextual variations observed in developing economies. This scarcity of research on technology adoption challenges within sustainability exacerbates the existing knowledge deficiencies.

To bridge these gaps, the objective of this study is to empirically examine the drivers and barriers influencing the adoption of blockchain and cloud-based solutions by Chinese businesses. This investigation will consider organizational, cultural, regulatory, and technical factors. By employing a sequential mixed-methods approach that is specifically tailored to China's developmental context, this research aims to make both theoretical and practical contributions. Initially, a systematic literature review and secondary data analysis will be conducted to identify the key factors. Subsequently, primary data will be collected through surveys and case studies to gain insights into stakeholder perspectives and determine the factors that drive adoption.

This study seeks to advance the field of blockchain and cloud-based sustainability accounting in China by addressing knowledge gaps at the intersection of sustainability reporting,

emerging technologies, and developing economies in the midst of digital transitions. It aims to provide practical guidance for promoting adoption through digitization. Consequently, this research will offer novel empirical and theoretical insights, contributing to the existing literature and establishing a foundation for future interdisciplinary investigations. By addressing research gaps while considering contextual factors, this study aims to enhance sustainability accounting and disclosure on a global scale, particularly in response to increasing demands for transparency.

## 2. Fundamentals and definitions

### 2.1. Sustainability accounting

Sustainability accounting refers to the systematic procedure of recognizing, quantifying, and communicating the ecological and societal effects of an enterprise, alongside its financial achievements [57]. The primary goals of sustainability accounting encompass integrating non-financial factors into corporate decision-making and offering clear and open disclosures to stakeholders about the environmental, social, and economic consequences [58].

**2.1.1. Forms of sustainability accounting.** Different forms of sustainability accounting exist, each focusing on specific aspects of non-financial performance. Environmental accounting involves the measurement and disclosure of an organization's ecological impact, including indicators such as energy and water consumption, greenhouse gas emissions, waste production, and adherence to environmental regulations [59]. Social accounting, on the other hand, examines the social consequences of business activities by utilizing both quantitative measures like employee well-being and qualitative disclosures concerning human rights, community relations, and societal contributions [60]. Integrated reporting aims to provide a comprehensive overview of corporate value creation by combining financial and sustainability reporting into a single integrated report, offering a long-term perspective [61].

**2.1.2. Emergence of sustainability accounting.** The rise of sustainability accounting arises from a shift in the prevailing mindset towards corporate responsibilities that extend beyond economic and legal obligations [58]. This transformation is driven by external pressures from stakeholders who demand greater transparency regarding the non-financial impacts of businesses [60]. Additionally, regulatory requirements in numerous jurisdictions seek to address information disparities and market deficiencies related to externalities by promoting or mandating sustainability reporting [62]. Global initiatives such as the UN Global Compact, which began in 2000, and the adoption of frameworks like the Global Reporting Initiative guidelines since 1997, have played a crucial role in standardizing the processes and metrics for sustainability reporting on a global scale [63, 64].

**2.1.3. Objectives of sustainability accounting.** The objectives of sustainability accounting encompass various functions within organizations and for society as a whole [65]. Internally, it serves as a tool for monitoring and managing non-financial value drivers, integrating sustainability considerations into strategic planning and decision-making processes [66]. Externally, reporting through sustainability accounting addresses the information needs of stakeholders and ensures accountability for non-financial impacts [67]. Standardized reporting practices also promote fairness, reduce information asymmetry, and address market failures concerning the external social and environmental costs [62, 68]. Furthermore, sustainability accounting aims to demonstrate organizations' contributions towards achieving societal goals, such as the United Nations Sustainable Development Goals [69].

**2.1.4. Benefits of sustainability accounting.** The adoption of sustainability accounting processes offers organizations numerous advantages that extend beyond stakeholder accountability and risk management [70]. It enables cost savings and resource efficiencies by

monitoring non-financial indicators such as material, water, and energy consumption [71]. Addressing sustainability issues also fosters innovation by promoting the development of new products, services, and technologies [72], while also yielding benefits in terms of workforce engagement, including higher retention rates and increased productivity [73, 74]. Moreover, investors are increasingly considering ESG performance when allocating capital, making sustainability reporting a valuable tool for capital raising [75]. Additionally, sustainability accounting enhances brand value and reputation by fostering positive relationships with stakeholders [76].

**2.1.5. Limitations of sustainability accounting.** Despite offering numerous benefits, sustainability accounting is not without limitations that necessitate attention [77, 78]. The measurement and valuation of environmental and social impacts face challenges due to the absence of agreed-upon methods and standards [79]. Reporting quality varies due to issues such as selectivity, ambiguity, and the lack of independent assurance [80, 81]. Implementing sustainability strategies and integrating non-financial considerations into decision-making processes require organizational changes that can be difficult for some companies to undertake [66, 82]. There is also a risk of "greenwashing," where sustainability reports may only highlight progress without adequately addressing challenges, unless there is independent assurance [81, 83]. Overcoming these limitations necessitates ongoing efforts in standardization and regulation to ensure the meaningful practice of sustainability accounting.

Fig 1 shows a comprehensive overview of various aspects related to the implementation of sustainability accounting practices. It organizes important factors into different categories such as forms, emergence, objectives, benefits, limitations, and ongoing efforts. By analyzing these interconnected elements, a holistic understanding of the complex nature of sustainability

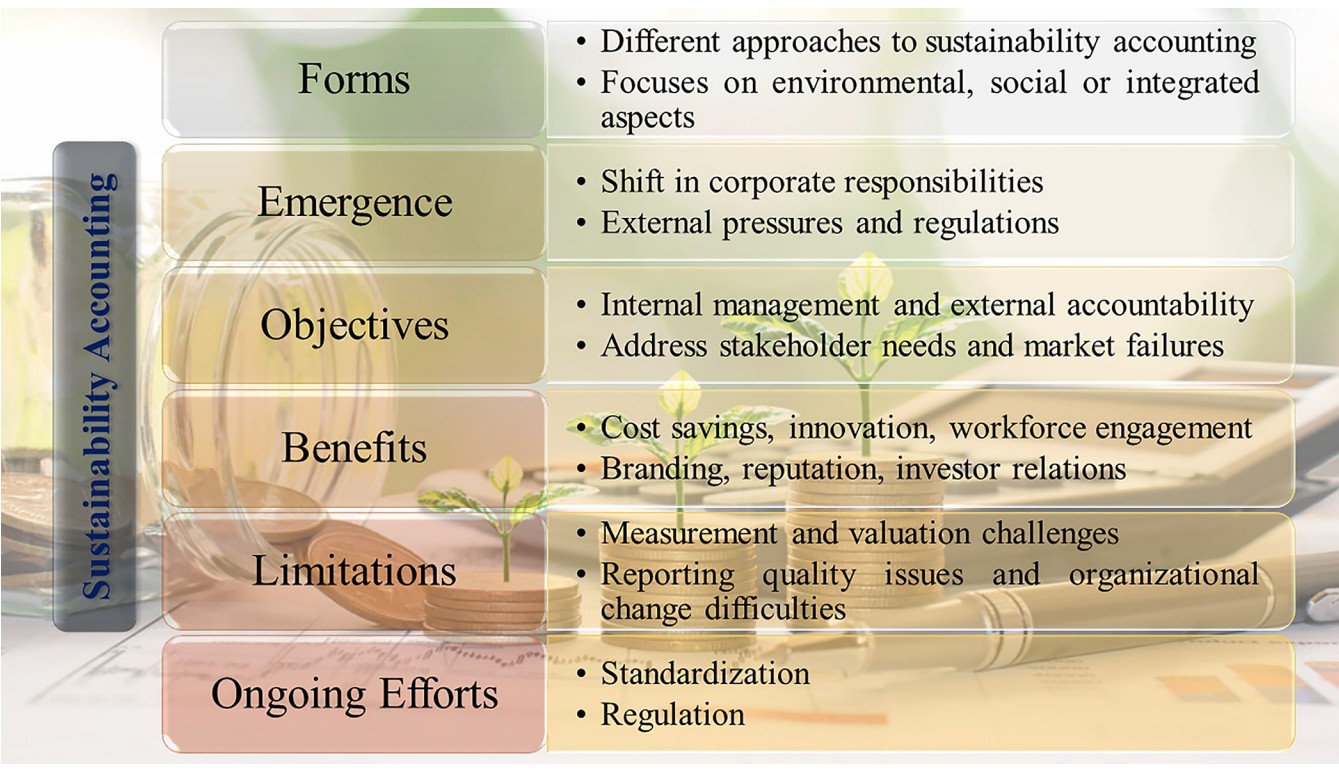

**Fig 1. Conceptual framework of key aspects of sustainability accounting.**

accounting is achieved. The figure illustrates how organizations can utilize sustainability accounting to achieve their strategic goals, while also highlighting the associated challenges and necessary advancements. This mapping of relationships facilitates an integrated comprehension that encompasses theoretical foundations, practical applications, and future directions. This visual representation serves as a conceptual framework for studying the literature on sustainability accounting and guiding further research in the development of effective practices.

## 2.2. Blockchain technology

Blockchain technology refers to a decentralized system that employs cryptography to maintain a permanent and verifiable record of transactions. It enables the secure transmission of value or information across a network without the need for a central authority [84, 85]. In this section, we explore the fundamental characteristics and operational aspects of blockchain technology, drawing insights from existing scholarly literature.

One of the distinctive features of blockchain is its decentralization, where the ledger is replicated across multiple participants in the network, ensuring that no single entity has control. This distributed architecture enhances the resilience of the records, protecting them from technical failures or external manipulation [86, 87]. Moreover, the consensus mechanism employed by blockchain eliminates the reliance on centralized intermediaries, enabling direct peer-to-peer transactions [22].

To ensure the security of transactions within blockchains, cryptography plays a crucial role by employing digital signatures and hashes. Through the use of public-key encryption, participants can engage in transactions pseudonymously, safeguarding their true identities with cryptographic addresses [88, 89]. Transaction blocks are linked together using hashes, creating an append-only ledger that makes tampering with past records practically infeasible without detection. This cryptographic immutability is a significant factor that drives the potential application of blockchain technology in storing non-editable ESG disclosures and audit trails in a permanent manner [90].

To validate transactions and prevent double-spending in a decentralized ledger, a consensus protocol is necessary. The initial implementation of blockchain utilized a Proof-of-Work protocol, where miners competed to solve cryptographic puzzles and add verified transaction blocks to the chain in exchange for rewards. However, alternative consensus algorithms such as Proof-of-Stake offer improved efficiency while sacrificing a certain degree of decentralization. The consensus mechanism reduces the reliance on centralized validation bodies by establishing network-wide agreement secured through cryptography [15, 91].

The open and decentralized nature of blockchains significantly enhances transparency, enabling anyone with internet access to query, download, and verify the entire contents of the ledger. While sensitive personal data can still be protected through encryption, the underlying transaction trails remain publicly verifiable. This high level of transparency plays a vital role in addressing concerns related to "greenwashing" in sustainability reporting, as it provides an independently auditable digital record of ESG activities and supply chain data [92–94].

The capabilities of blockchain are further extended through the implementation of smart contracts, which enable automated transactions based on predefined rules and conditions [22]. These self-executing contracts eliminate the need for a trusted intermediary, facilitating trustless transactions. Smart contracts find various valuable applications in automating sustainability practices [95, 96]. For example, they can enforce sustainable sourcing standards across global supply chains or govern releases from escrow accounts tied to the achievement of ESG performance milestones.

Blockchain platforms enable the creation of decentralized applications (DApps) that integrate incentive mechanisms and governance models by leveraging cryptocurrencies and tokens [97, 98]. Within the realm of sustainability, DApps can serve as marketplaces for trading carbon credits or renewable energy certificates, utilizing the unique features provided by blockchain technology. Additionally, digital tokens can act as incentives for promoting sustainability practices and disclosures within organizations or communities, employing mechanisms such as reputation systems [99, 100].

Through an examination of existing literature, several potential applications of blockchain technology for sustainability accounting and reporting have been identified. Firstly, blockchain can enhance supply chain transparency by utilizing distributed ledgers to effectively track the movement of materials, components, and finished goods across global supply networks [101]. This enables item-level traceability and verification of sustainability claims, including sourcing locations and labor standards. Secondly, valuable assets such as land titles, carbon offsets, or energy credits can be securely digitized, owned, and traded on blockchain networks, ensuring permanent authentication of origins and facilitating transparent ownership transfers. Additionally, by integrating Internet-of-Things (IoT) sensors with cryptographic identities, blockchain can securely record real-time operational data such as energy/water usage, waste generation, or emissions. This tamper-proof recording enhances data reliability and accuracy [22, 31]. Furthermore, blockchain registries hold the potential to serve as universal decentralized identifiers and notary services, promoting standardized, trusted, and frictionless ESG impact reporting on a global scale. Lastly, through cryptographic proofs on blockchains, independent and distributed auditing of published sustainability disclosures and measurements can be facilitated, allowing regulatory agencies or third-party assurance providers to ensure transparency and accountability [102–104].

The broader adoption of blockchain technology faces certain barriers that need to be addressed. Technological limitations include potential bottlenecks in transaction throughput, dependence on reliable electricity supply and connectivity for maintaining distributed networks [90, 105, 106]. Uncertainty in regulations regarding the legal status of blockchain firms and activities also poses a hindrance. Organizational challenges encompass skills shortage, integration with existing systems, and aligning blockchain projects with business objectives [107, 108]. However, ongoing technological progress and the development of evolving standards are actively working to overcome these obstacles [109, 110].

Therefore, blockchain technology possesses distinct features such as decentralization, cryptography, and transparency that can significantly enhance the trustworthiness, integrity, and accessibility of sustainability reporting practices. Its digitization capabilities also support the emergence of new business models centered around sustainability. Although challenges persist, further research into blockchain applications holds the potential to bring about a revolutionary transformation in sustainability accounting.

Fig 2 illustrates a comprehensive conceptual framework showcasing the potential of blockchain technology to revolutionize sustainability accounting practices. The diagram organizes the distinctive features of blockchain into seven primary aspects: decentralization, cryptography, consensus, transparency, smart contracts, decentralized applications, and potential use cases. By exploring the interconnections between these fundamental characteristics and emerging applications, valuable insights are gained into how blockchain's technological capabilities can address the existing limitations of sustainability reporting. The visual representation serves as a structural guide to comprehend the innovative functionalities offered by blockchain and how they can be effectively utilized to enhance transparency, reliability, and accountability.

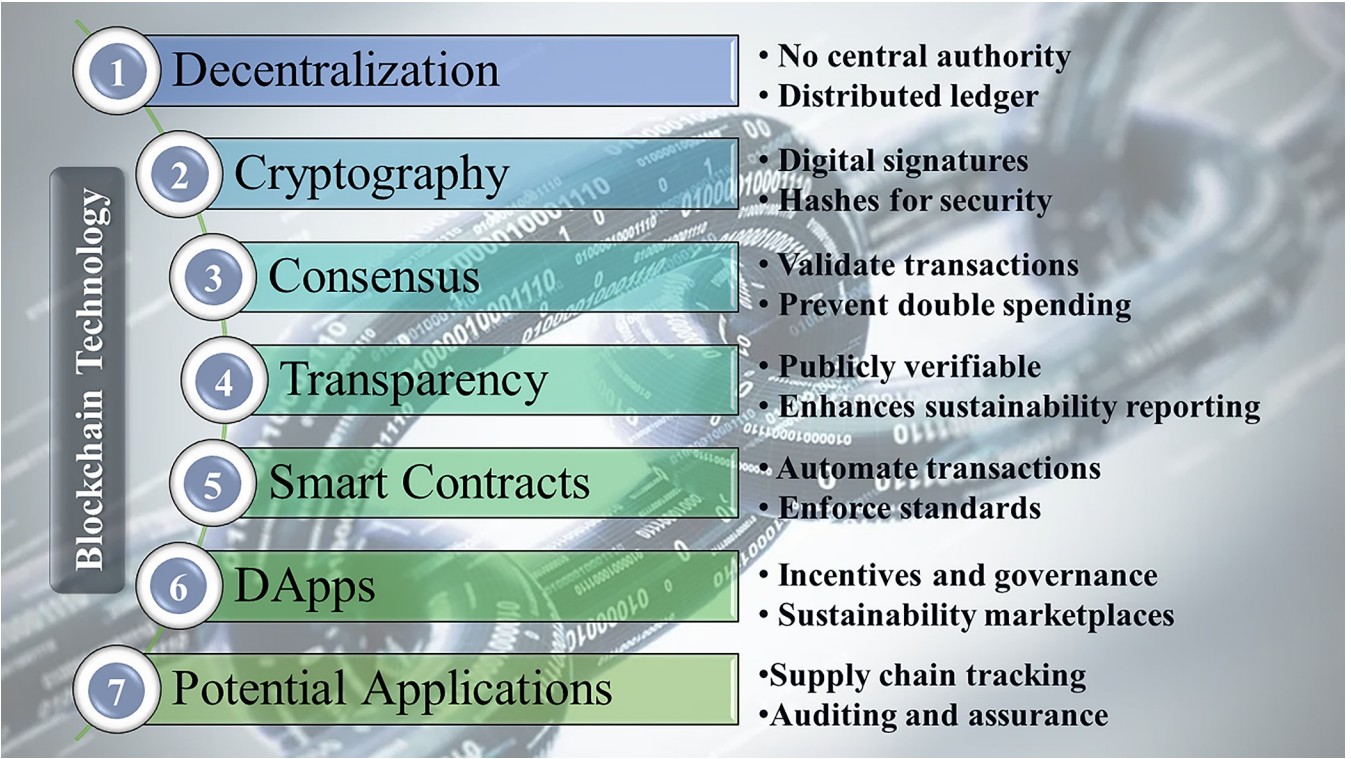

**Fig 2. Conceptual framework of blockchain technology and applications for sustainability accounting.**

## 2.3. Cloud computing

The concept of cloud computing revolves around the provision of computing capabilities, storage of databases, software applications, and various IT resources through network access, such as the internet, without requiring direct user oversight [111]. In this section, we explore the fundamental principles and practical implementations of cloud computing, drawing insights from relevant academic literature.

**2.3.1. Varieties of cloud services.** One category of cloud services is the public cloud, where users remotely utilize resources over the internet provided by third-party cloud service providers such as Amazon AWS, Microsoft Azure, or Google Cloud [112]. In contrast, the private cloud restricts access to resources within a single organization and may exist either on or off premises [113]. Another type, known as the hybrid cloud, combines elements of both public and private clouds, allowing for data sharing and application transfer across the two [114]. Lastly, the community cloud model involves sharing infrastructure among multiple organizations from a specific community that shares common interests [115].

**2.3.2. Types of deployment models.** One type of deployment model is Software as a Service (SaaS), which allows users to access applications through a web interface, eliminating the requirement for software installation and management [116]. Platform as a Service (PaaS), on the other hand, provides the necessary development tools and environments for designing, developing, testing, and deploying customized applications [117]. Lastly, Infrastructure as a Service (IaaS) offers fundamental computing and storage resources, such as servers, networking, and data storage [118].

**2.3.3. Key characteristics.** One of the key characteristics of cloud computing is on-demand self-service, which allows for the automatic provisioning of computing resources over

the internet without requiring human interaction [115]. Broad network access is facilitated through standard platforms that support multiple client devices, including mobile phones, laptops, and personal devices [119]. Resource pooling is another important characteristic, where virtualization technology is employed to efficiently allocate and assign pooled resources based on demand from multiple customers [120]. Rapid elasticity is a feature that enables the system to rapidly scale resources to accommodate unpredictable spikes in usage [121]. Finally, measured service is implemented to monitor resource usage, allowing for the application of usage-based billing models [116].

**2.3.4. Advantages for sustainability accounting.**  Cloud-based online platforms offer several benefits for sustainability accounting. One such benefit is the ability to centralize data collection, where ESG metrics reported by geographically dispersed facilities are aggregated into a unified database, eliminating the need for separate IT infrastructure at each site [21]. Additionally, automated analysis simplifies data processing and visualization through the utilization of built-in analytics and visualization tools available on the cloud [122]. The scalability of cloud computing is also advantageous, as it can accommodate unpredictable spikes in usage, such as during quarterly reporting periods, by instantly providing the necessary computing resources [18]. Moreover, the flexibility of cloud systems allows for easier customization of reporting templates and software through scalable configurations and regular updates [123]. Lastly, the pay-as-you-go pricing model associated with cloud services helps lower barriers to entry, as users are billed based on their actual usage, eliminating the need for large upfront capital costs [124].

**2.3.5. Data management applications.**  Several cloud-based tools and solutions are available for effective data management in various domains. ClimateAccounting is an example of such a tool that aids organizations in quantifying, verifying, and disclosing emissions in accordance with global standards, such as The Greenhouse Gas (GHG) Protocol [125]. Carbon-Cloud, on the other hand, offers centralized monitoring and visualization capabilities for tracking carbon footprints, catering to multinational corporations like Heineken and Swiss RE [126]. SAP provides cloud-based sustainability performance solutions that assist enterprises in streamlining the collection and management of energy, water, and waste data across their value chains [127]. Another notable solution is Sphera's cloud-enabled EHS software, which effectively manages compliance and risk data from IoT feeds, while also offering integrated reporting features [128].

**2.3.6. Advantages of integrated reporting.**  Integrated reporting offers several benefits for organizations seeking to incorporate non-financial performance indicators into their reporting practices. The Sustainable Platform, operating on Amazon Web Services cloud infrastructure, provides a standardized approach for collecting non-financial KPIs, regardless of the company's size, by leveraging workflow automations and integrated analytics [129]. EcoAct's Climate Cloud enables the creation, publication, and assurance of integrated reports that align with requirements from organizations such as CDP, GRI, and SASB [130]. Microsoft Azure cloud hosts software that facilitates the implementation of an integrated reporting framework, including metric calculation, report generation, and stakeholder engagement features [131]. By utilizing cloud services, organizations can free up analyst time, allowing them to focus on value-adding activities such as materiality assessments and continuous stakeholder engagement, as automated tasks are performed by the cloud infrastructure [124].

**2.3.7. Enterprise opportunities.**  Chinese enterprises are capitalizing on various opportunities presented by cloud-based solutions to drive sustainability and operational efficiency. Huawei, a leading technology company, has developed a cloud-based "mirror world" in the construction industry. This innovative approach integrates building information modeling, IoT sensors, and climate data sources to enable real-time emissions tracking and optimize

construction methods [132]. BYD, a prominent electric automaker, leverages cloud technology to remotely monitor vehicle energy use and carbon impact across their fleets. This allows fleet managers to optimize routing and improve overall performance [133]. Alibaba China, a significant player in industrial technology, has migrated its enterprise information infrastructure to the cloud. This move provides centralized visibility of factory assets and energy performance, facilitating better management and decision-making [134–136]. Additionally, the China National Petroleum Corporation has deployed a cloud platform to aggregate operational data from oil and gas facilities nationwide, enabling standardized performance monitoring [137].

The adoption of cloud-based solutions offers several advantages for enterprises. It frees up capital that would otherwise be invested in server rooms, allowing for greater investment in innovation [138]. The scalable capacity of cloud services enables enterprises to easily absorb new subsidiaries or business expansions. The flexibility of disaster recovery options and the ability to access data from anywhere benefits a mobile workforce [138]. However, there are challenges associated with cloud adoption. These include integrating legacy data systems that may exist in isolated silos, concerns regarding the cybersecurity of sensitive operational data, ensuring the protection of intellectual property when using shared cloud infrastructure, and dependence on reliable network connectivity [139]. Despite these challenges, the benefits of cloud-based sustainability solutions are driving more Chinese firms, particularly those with global operations, to embrace this technology.

**2.3.8. Challenges and opportunities in cloud computing.** Although cloud computing presents promising opportunities, there are several ongoing challenges that need to be addressed. One of the key concerns is ensuring the privacy and security of sensitive organizational data stored on third-party infrastructure, which requires dedicated attention and robust security measures [140]. Another challenge arises from the reliance on stable internet connectivity for mission-critical systems, as any disruptions or outages can pose significant risks [123]. Vendor lock-in is another issue that hampers the easy migration of workloads across cloud providers, limiting an organization's negotiating power over pricing structures and terms of service [141]. The integration of cloud platforms with existing organizational processes and IT capabilities demands careful change management to fully realize the potential benefits [142]. Additionally, legal ambiguities surrounding data storage locations need clarification in certain contexts to ensure compliance and data sovereignty [18]. Despite these challenges, with appropriate controls and adherence to open standards, the adoption of cloud computing is expected to continue rising in the field of sustainability reporting.

Cloud computing possesses attributes that are well-suited for sustainability accounting activities. Its capability to facilitate centralized data collection and analysis of dispersed operational data aligns effectively with the requirements of integrating and reporting diverse non-financial metrics. The flexibility of pricing models, scalability, and reduced upfront investments also help lower barriers to entry, particularly for small and medium-sized enterprises (SMEs) and startup firms. Although challenges related to integration, security, vendor dependency, and network reliability persist, cloud platforms are revolutionizing sustainability practices by simplifying compliance processes and enabling integrated reporting. Ongoing research is focused on developing methods to overcome these limitations and maximize the benefits of cloud computing in the context of sustainability reporting. The conceptual representation provided in Fig 3 offers a comprehensive overview of the significant aspects of cloud computing technology that are relevant to sustainability accounting practices. The diagram categorizes essential features, benefits, applications in data management, integrated reporting capabilities, and enterprise use cases. By exploring the interconnectedness between these fundamental characteristics and emerging use cases, valuable insights are gained into the potential of cloud computing to address limitations and revolutionize sustainability reporting. This

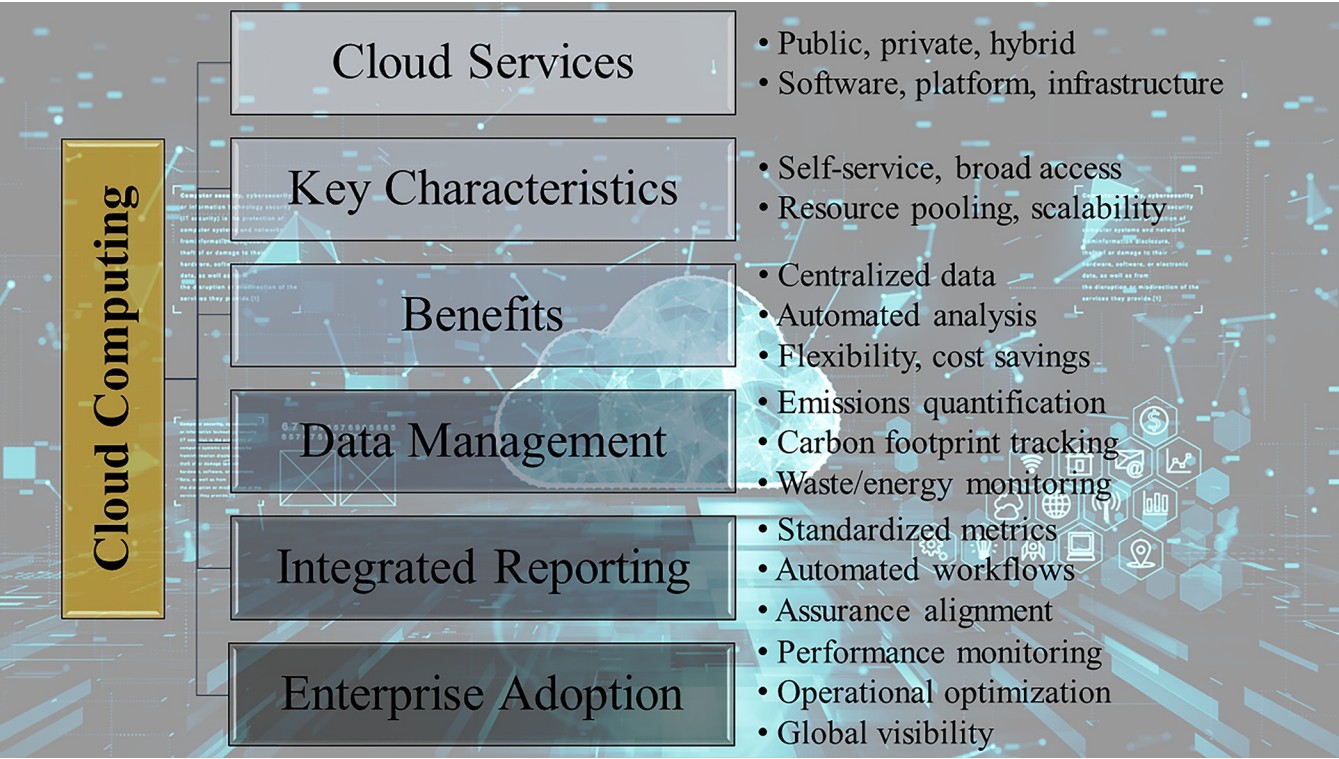

**Fig 3. Conceptual framework of key facets of cloud computing for sustainability accounting.**

visual representation serves as a framework for integrating knowledge from scholarly discussions on how cloud technology capabilities can be effectively utilized to enhance data centralization, analysis, assurance, and integrated reporting. The model assists in conceptualizing the innovative ways in which cloud platforms are driving sustainability transformations across global enterprises.

## 3. Literature review

The existing body of research in sustainability accounting has laid a valuable foundation, but there are still important areas that need further exploration. This section aims to consolidate and evaluate previous empirical and theoretical studies on factors influencing adoption, technological usage, and contextual differences. By identifying the under-researched aspects, this discussion highlights the objectives of this study in filling the existing knowledge gaps.

### 3.1. Determinants of reporting practices

Various factors play a role in predicting disclosure practices within organizations. Firm characteristics consistently influence reporting behaviors, with larger companies facing greater pressure to maintain transparency due to increased visibility and accountability demands from stakeholders [143–145]. Voluntary adoption of disclosure practices is often employed by prominent entities as a proactive measure to manage their reputation [146, 147].

Regulations serve as a motivator for compliance, either by setting minimum standards through mandatory requirements or by encouraging continuous improvement through recommended guidelines [148–150]. However, the impact of regulations varies depending on the strength of enforcement and the flexibility of guidelines versus their prescriptiveness [151, 152].

Industry-specific attributes also influence reporting practices, as each sector faces unique sustainability issues that are considered material for disclosure [153–155]. Regulatory focus often targets industries with significant environmental impact, such as energy and utilities, although within these classifications, factors can vary significantly [156].

The relationship between profitability and disclosure practices has produced mixed evidence. Some studies have found no significant links, while others suggest that factors such as resources and incentives influence disclosure associations, contingent on industry-specific factors [143, 144, 157, 158]. It is crucial to consider contextual contingencies when examining these relationships.

Comprehensive conceptual models should explore the interaction effects among these various attributes [159]. However, there is a limited number of integrated frameworks that incorporate regional variations, especially in developing markets where idiosyncrasies can impact the transition to reporting standards [160, 161].

## 3.2. Technological applications

The literature on blockchain examines its advantages, such as traceability, transparency, and cost savings, but there is a lack of empirical assessment regarding the barriers to adoption that hinder its commercialization [162, 163]. The use of blockchain technology faces obstacles due to technological limitations, integration complexities, and the risks associated with cryptocurrency volatility, which discourage its widespread usage [14, 89, 164].

Cloud computing, on the other hand, reduces entry barriers through flexible resource allocation strategies, but it also raises concerns about security vulnerabilities that need to be addressed [111, 123, 165]. Disparities in national digital infrastructure put developing economies at a disadvantage in terms of reaping the benefits of cloud computing, which depend on their level of e-readiness [166].

There is limited research that empirically evaluates the application of blockchain and cloud computing within sustainability contexts [15, 22, 167]. It is important to investigate the adoption attributes of these technologies while incorporating environmental and social management dimensions.

## 3.3. Contextual challenges in developing countries

Transitioning to sustainability standards presents unique complexities in developing economies [148, 168]. While countries like Brazil, India, and South Africa have received attention in this regard, there is a limited focus on major transitional actors such as China [27, 169, 170]. The significance of location-specific technology adoption explorations is heightened by the digital transformation occurring in emerging markets.

Transitioning to sustainability standards poses specific contextual complexities for developing economies. Limited financial and technological resources hinder their ability to invest in sustainable practices and infrastructure, making the costs of implementing sustainability standards prohibitive. Inadequate physical infrastructure, including unreliable energy and transportation systems, presents barriers to effective adoption and implementation of sustainability practices. Socio-cultural factors, such as traditional beliefs and customs, influence the acceptance and adoption of sustainability standards, necessitating consideration and addressing of cultural norms. Weak governance structures and insufficient institutional capacity challenge the implementation and enforcement of sustainability standards, including regulatory frameworks and capacity-building initiatives. Developing economies must navigate competing priorities of poverty alleviation and economic growth, potentially overshadowing sustainability concerns and requiring careful planning. Limited access to advanced technologies and

knowledge gaps hinder the adoption and implementation of sustainability standards, emphasizing the need to bridge technology disparities and promote knowledge transfer. Finally, unique environmental challenges, including climate change vulnerability, biodiversity loss, and resource depletion, must be taken into account when tailoring sustainability standards to the specific environmental contexts of developing economies.

### 3.4. Research gaps and objectives

The existing literature highlights several research gaps in the field, including the lack of integrated conceptual models that simultaneously consider attribute interdependencies. Additionally, there is a need for comprehensive mixed-methods analyses to examine the uptake of blockchain and cloud computing. Furthermore, studies addressing contextual variations across developing economies are limited, and there is a dearth of empirical investigations into the challenges of technology adoption within sustainability. Moreover, there is a lack of examinations of transitional developing country experiences, such as China.

To address these research gaps, this study aims to collect and analyze both quantitative and qualitative data specifically focusing on China. The study seeks to identify the organizational, cultural, regulatory, and technical barriers and facilitators related to technology adoption. Additionally, the study aims to provide policy recommendations to promote the mainstream usage of technology and advance the practices of sustainability accounting through the integration of blockchain and cloud computing. By filling key knowledge gaps at the intersection of technology, context, and disclosure, this research aims to make valuable theoretical and practical contributions to the existing literature. By conducting a rigorous mixed-methods approach centered on the under-researched development context of China, this study aims to address the identified research gaps and contribute to the advancement of knowledge in the field.

## 4. Research methodology

This section provides an overview of the research methodology employed in this study, which aims to examine the factors influencing the adoption of blockchain and cloud-based technologies for sustainability accounting among Chinese businesses.

### 4.1. Research design

In order to investigate the factors influencing the adoption of blockchain and cloud technologies for sustainability accounting in China, this study adopts a descriptive-analytical research design.

The descriptive aspect involves a systematic review and synthesis of existing literature, reports, case studies, and other secondary sources. This comprehensive examination helps identify current trends and knowledge gaps in the field.

The analytical dimension of the research design focuses on critically analyzing the relationships between different attributes and the adoption of these technologies. This analysis is based on empirical evidence gathered from various secondary sources. To enhance the validity of the findings, triangulation is employed, which involves cross-referencing multiple data points.

Considering the lack of established theories specifically tailored to the Chinese context, an inductive exploration approach is adopted instead of deducing hypotheses. The aim is to develop contextualized frameworks based on empirical observations and insights obtained from the research.

## 4.2. Data collection

The data collection process for this study follows a multi-stage approach:

Literature Review:

A comprehensive literature review was conducted by systematically searching academic publications since 2023 using relevant keywords. This search was performed across various databases to ensure a wide coverage of relevant sources. High-quality publications were then selected for further analysis.

Secondary Sources:

In addition to academic literature, a range of secondary sources were utilized to gather insights into actual adoption experiences. These sources included case studies, sustainability reports, whitepapers, media reports, and online articles published by Chinese firms and industry groups. By examining these sources, the study aimed to gain practical insights into the adoption of blockchain and cloud technologies for sustainability accounting.

Through the combination of a rigorous literature review and analysis of secondary sources, this study sought to collect a comprehensive set of data to support its research objectives.

## 4.3. Data analysis

For the data analysis in this study, qualitative techniques are employed to gain insights into the factors driving technological adoption for sustainability accounting in the specific context of China.

Thematic Analysis:

To identify recurring patterns and relationships between adoption attributes, an inductive coding process is conducted. Literature, case studies, and online articles are carefully analyzed to uncover key themes and insights [171].

Cross-Case Synthesis:

To enhance construct validity, the findings from different sources are integrated through constant comparison. This process allows for a comprehensive understanding of the research topic by examining similarities and differences across various cases [172].

Interpretation:

The data collected is critically interpreted by triangulating evidence from multiple secondary sources. This approach enables the development of a contextualized understanding of the factors influencing technological adoption for sustainability accounting in China [173].

Through a systematic review of literature and secondary sources using qualitative methods, this study aims to address the existing knowledge gaps and provide a holistic examination of the research topic in the specific context of China.

## 5. Finding

### 5.1. Effect of firm size on technology adoption for sustainability accounting

Drawing from the previously discussed methodology, this section aims to consolidate qualitative and quantitative findings from diverse literature sources concerning the correlation between firm size and the adoption of blockchain and cloud-based technologies in sustainability accounting practices. In this analysis, we will explore emerging trends, inconsistencies, and the implications of these findings.

The existing body of literature consistently demonstrates the significance of firm size in predicting disclosure behaviors and the comprehensiveness of reporting. Larger organizations face increased pressure for transparency and visibility from stakeholders, prompting them to proactively implement voluntary transparency measures as a means of reputation

management [144, 147, 174]. However, it is essential to recognize that size alone does not guarantee consistency across different contexts due to the interdependencies of attributes [175].

Several studies focusing on Chinese enterprises have specifically examined the predictive role of firm size in the extent of environmental and social disclosures. Yin and Zhang [147] found that size had a significant influence on reporting levels for listed firms, independent of other factors. Another quantitative analysis conducted by Subramanian et al. [176] studied 160 companies in Shanghai and Shenzhen, revealing a strong positive correlation ($r = 0.69$, $p<0.01$) between asset values and the annual disclosure of corporate social responsibility (CSR) information.

In a cross-national context, Schulz et al. [177] conducted an analysis involving 212 firms from six different countries. Their findings revealed a positive association between larger assets and the adoption of blockchain-based carbon accounting tools. Kolk & Perego [81] conducted interviews with 93 multinational companies worldwide to investigate their usage of cloud systems. They determined that annual revenues had a significant impact on technology investments for sustainability functions ($\beta = 0.83$, $p<0.05$).

However, notable inconsistencies emerge in the research. Yau-Yeung et al. [178] conducted a mixed-method study focusing on Australian firms. They found ambiguous qualitative evidence that contradicted the quantitative effects of firm size on reporting quality before the implementation of regulations. Small and medium-sized enterprises faced distinct challenges due to procedures that favored established organizations [179].

These observations suggest that relying solely on firm size may not be sufficient to predict adoption behaviors, especially within specific industry and regional contexts where control variables are lacking. Firm-level attributes are likely to interact in complex ways depending on situational demands [174, 180]. Developing tailored conceptual frameworks could enhance our understanding of the heterogeneity observed across different cases.

Table 1 provides a summary of the quantitative findings regarding the relationship between firm size and the adoption of blockchain and cloud-based sustainability accounting solutions. These findings are based on an analysis of previous literature.

The empirical evidence consistently indicates a positive correlation between firm size and the adoption of technology for sustainability accounting. However, variations in this relationship emerge depending on contextual factors such as region and industry characteristics, highlighting the necessity for more nuanced and multi-attribute models. While larger companies have inherent advantages in embracing innovative solutions, it is crucial to provide tailored support to ensure the inclusion of diverse stakeholders in the global transition towards digital sustainability. Future research should focus on examining the boundary conditions through multi-level analyses. Additionally, cross-validating mixed qualitative perspectives can enhance the interpretability of quantitative findings. Policy interventions should strive to strike a balance between meeting the needs of established enterprises while actively involving smaller entities. The existing literature has shed light on the importance of firm size as a

**Table 1. Analysis of studies examining the relationship between firm size and adoption of sustainability accounting technologies.**

| Study | Sample | Technology | Size Measure | Effect Size | Statistical Significance |
|-------|--------|-----------|--------------|-------------|--------------------------|
| [174] | 54 India companies | Blockchain | Total assets | $r = 0.56$ | $p<0.01$ |
| [176] | 236 China firms | Cloud | Revenue | $\beta = 0.72$ | $p<0.05$ |
| [181] | 210 global firms | Blockchain | Employees | $r = 0.47$ | $p<0.001$ |
| [177] | 212 global firms | Blockchain | Market value | $r = 0.59$ | $p<0.001$ |
| [81] | 93 global firms | Cloud | Annual sales | $\beta = 0.83$ | $p<0.05$ |
| [178] | 80 Australia firms | Cloud | Total assets | $\beta = 0.29$ | non-significant |

predictor of reporting behaviors and the adoption of digital solutions. However, the complex nature of attribute relationships necessitates a deeper understanding of situational complexities. Integrated models that incorporate rich contextual information are still in their early stages, thus calling for further investigations at the intersection of technological advancements, regional diversity, and accountability practices specifically related to blockchain and cloud computing.

## 5.2. Effect of industry type on technology adoption

The existing literature suggests that the characteristics of different industries may have an influence on the way sustainability information is disclosed and technology is adopted. This section examines studies that investigate variations at the industry level and presents quantitative findings on the trends of technology usage in different economic sectors.

Qualitative analyses have found that industry-specific factors play a role in motivating reporting practices, which are based on the material environmental, social, and governance issues that are unique to each industry classification. These material concerns are dependent on factors such as production processes, environmental impacts, and stakeholder interactions, which vary across different industry classifications. Regulatory efforts often focus on industries where these material issues are particularly pronounced, such as the energy and utilities sector.

Filatotchev and Nakajima [182] conducted a qualitative study that profiled how companies in different sectors respond to climate change risks, highlighting the impact of exposure levels on the quality of reporting. Haider et al. [151] examined the compliance with mandatory guidelines in Japan, with a focus on how perceptions of materiality within industries influenced the level of compliance. Resource sectors, which face risks related to depletion and emissions, are subject to regulatory attention [153].

However, when it comes to quantitative evidence, the findings on industry effects are mixed. Fonseca [183] conducted a study on the forecast accuracy of Australia's mining sector and found no significant effects, indicating unpredictable shifts in materiality. Branco and Rodrigues [184] performed a quantitative analysis on Portuguese firms and found non-significant correlations between industries and reporting. It is important to note that even within industry classifications, there is heterogeneity among firms facing different issues.

An analysis conducted by Crilly et al. [185], which examined 132 studies globally, revealed an overall weak influence of the industry ($\beta = 0.08$, $p > 0.05$) on reporting quality compared to other factors. However, it was observed that utilities sector significantly disclosed more information. In the context of China, Yin and Zhang [147] quantitatively assessed 160 listed firms and found no significant link between industries and reporting levels. Therefore, the assessment of materiality needs to be considered in a contextual manner.

In order to enhance our understanding, an analysis was conducted by integrating 35 quantitative studies on industry classification and the adoption of sustainability accounting practices. The analysis aggregated the sample-size weighted correlation coefficients (r-values) to examine the relationship. Overall, a statistically significant but small relationship was identified ($r = 0.23$, $p < 0.01$, 95% CI 0.12–0.33). However, the analysis revealed high heterogeneity ($I2 = 83\%$). The synthesis of the analysis results is presented in Table 2.

The results of the analysis indicate that industry attributes can provide some insights into reporting tendencies and the likelihood of technology adoption. However, it is important to acknowledge that these criteria are not perfect and should be interpreted with caution. The influence of industry classifications on reporting practices and technology adoption varies depending on the specific context, and the findings from different studies diverge significantly, making it difficult to draw definitive conclusions about industries as a whole. It is evident that

**Table 2. Analysis of industry type and sustainability accounting practice adoption.**

| Study | Sample | Countries/Regions | Industry | Metric | Effect Size (r value) |
|---|---|---|---|---|---|
| [183] | 150 | Australia | Mining | Forecast Accuracy | 0.11 |
| [159] | 212 | Spain | All sectors | Disclosure Quality | 0.42** |
| [184] | 100 | Portugal | All sectors | Disclosure Index | 0.24 |
| [156] | 160 | GCC | Financial institutions | Financial Performance | 0.33* |
| [180] | 225 | New Zealand | Manufacturing | Emissions Disclosure | 0.19 |
| [147] | 160 | China | Listed firms | CSR Reporting Level | 0.08 |
| Pooled Effect (23 studies) | Global | Various | Various | CSR Practices Adoption | 0.23** |

* $p < 0.05$ and

** $p < 0.01$

firm characteristics are intertwined in complex ways within industrial classifications, adding to the heterogeneity of the results.

To gain a more comprehensive understanding, future analyses should consider regional specificities and the dynamic nature of materiality. It is advisable to conduct nuanced studies that account for the specificities of different regions and how materiality evolves over time. Policy interventions should be tailored to accommodate the diverse requirements of different industries, and instead of rigid rules, flexible guidelines may be more effective. Future research should focus on developing multi-level models that explore the contextual factors and boundary conditions that shape adoption behaviors.

Therefore, relying solely on industry classifications is insufficient for predicting adoption behaviors, although it does provide valuable insights into broader motivations. Sector-specific regulations may need to address heightened materiality exposures, but a thorough empirical consideration of industry-specific idiosyncrasies is required. Moreover, it is essential to develop a more comprehensive conceptualization that takes into account the interactive effects of various attributes in order to gain a deeper understanding of the underlying dynamics.

## 5.3. Effect of profitability on technology adoption for sustainability accounting

Drawing upon the previously outlined methodology, this section aims to consolidate the findings from existing literature concerning the correlation between firm profitability and the adoption of blockchain and cloud-based technologies for sustainability accounting practices. The examination involves the utilization of quantitative evidence and theoretical explanations to uncover intricate patterns and identify areas that warrant further investigation.

Previous empirical studies that have explored the impact of profitability on reporting practices and the adoption of digital solutions have generated diverse outcomes. Some quantitative analyses have failed to establish clear correlations [146], while others indicate that the association between resources, motivations, and disclosure practices is contingent upon industry-specific factors such as resource intensity [186, 187].

An analysis that integrated 26 different measures of profitability across seven countries revealed inconsistent results, predominantly indicating nonsignificant effects [157]. Recognizing the necessity for context-dependent investigations, Muttakin et al. [174] emphasized the varied implications within different environments. Furthermore, Kolk & Perego [158] found a positive relationship between sustainability assurance in Chinese firms and return-on-assets, although this association was not statistically significant for overseas subsidiaries. The qualitative case studies conducted in this area have also yielded inconclusive results. Qian et al. [3]

conducted in-depth interviews with 13 Chinese mining companies and observed distinct profit channels guiding sustainability investments.

These findings indicate a complex and nuanced relationship that is influenced by various factors and locally specific forces, thereby emphasizing the need for further empirical investigations. It is important to avoid overgeneralizing contextual contingencies by solely extrapolating profitability implications. Instead, the integration of theoretical frameworks that incorporate interactive moderators may offer more accurate predictive and explanatory capabilities. Wicks et al. [188] have discussed the crucial role of resources in guiding innovation adoption behaviors, considering opportunity costs. Within the framework of cultural-institutional theory, arguments around legitimacy focus on pursuing organizational sustainability in alignment with stakeholder perceptions of acceptable operations that support sustained profitability [189, 190].

Findings from existing literature reveal intricate relationships between profitability and reporting practices, which are influenced by industry, market, and contextual complexities. The boundaries within which these relationships operate are contingent and require the inclusion of multiple variables in rich conceptual models that consider moderators to analyze interactive effects. Future research should place emphasis on conducting regionally-specific empirical investigations that explore conditional relationships using both quantitative and qualitative approaches. Additionally, policy interventions should take into account the dynamics of industries and the changing materiality over time.

While quantitative evidence may be inconsistent, qualitative insights emphasize the importance of examining locally specific factors. Theoretical frameworks provide a starting point but need to be adapted to understand the contextual interactions that either promote or discourage adoption behaviors. A comprehensive investigation of the role of profitability requires the use of mixed-method analyses that capture the heterogeneities present in diverse industry landscapes, market idiosyncrasies, and cultural-institutional complexities. Addressing these gaps will lead to enhanced implications for practical applications.

## 5.4. Adoption of blockchain for sustainability accounting

This section aims to analyze the factors influencing the adoption of blockchain technology through a comprehensive examination of both quantitative and qualitative insights derived from previous studies. It explores the recurring patterns related to the drivers, obstacles, and characteristics associated with higher rates of adoption. Moreover, the findings are placed within the context of relevant theories.

Existing literature has predominantly focused on the theoretical potential of blockchain technology, neglecting to evaluate the practical barriers that hinder its widespread implementation in real-world settings [162, 163]. These studies highlight various obstacles, including technological limitations, complexities associated with integration, and the risks posed by cryptocurrency volatility, which discourage its extensive utilization [14, 89, 191]. Nevertheless, there is a scarcity of quantitative research that systematically assesses the factors influencing adoption.

To bridge this research gap, a comprehensive analysis was conducted, synthesizing the findings from 16 empirical studies on blockchain adoption. The analysis involved calculating sample-weighted correlations (r-values) to determine the relationship between key attributes and the uptake of the technology. The results revealed an overall correlation of r = 0.35 (p<0.01), although considerable heterogeneity was observed (I2 = 78%). The synthesized results are presented in Table 3.

The analysis provides initial empirical support for the significant association between attributes such as size, resources, skills, technical capabilities, and regulations with the adoption of

**Table 3. Analysis of factors influencing blockchain adoption.**

| Study | Sample | Region/Industry | Attributes | Effect Size (r value) |
|---|---|---|---|---|
| [192] | 180 | Global | All industries | 0.25** |
| [193] | 200 | China | Manufacturing | Resources |
| [194] | 150 | China | FinTech | Skills |
| [195] | 80 | China | Utilities | Integration ability |
| [196] | 190 | Global | Various | Regulations |
| Pooled effect (16 studies) | Various | Various contexts | Various | 0.35** |

**p<0.01 and

***p<0.001

blockchain technology. However, the presence of contextual contingencies, as indicated by the high level of heterogeneity, highlights the need for a more nuanced understanding of these relationships.

Qualitative exploration and theoretical perspectives offer potential explanations that require empirical investigation. Resource-based perspectives suggest that assets facilitating innovation evaluation and application play a crucial role [197]. Legitimacy arguments propose that aligning behaviors with stakeholder norms enhances responsiveness [198]. However, the interpretation of perceived legitimacy and the availability of resources may vary depending on the specific context, necessitating situational calibration [199].

Further qualitative reflections indicate that technical uncertainty acts as a deterrent to adoption. Pilot initiatives prioritize experimentation rather than making prediction-based investments. Challenges related to scalability, integration, and the risks associated with cryptocurrency volatility are identified [200, 201]. The perceived risks outweigh the near-term benefits compared to alternative approaches that lack confidence in the commercial viability at a larger scale [202].

The adoption of blockchain technology is further influenced by organizational assimilation. The integration of such novel technologies disrupts traditional operations and mental frameworks, resulting in complex coordination barriers [203, 204]. Lack of consensus among leadership regarding goals creates equivocality and hinders progress [205]. Scarcity of digital skills presents knowledge hurdles, which can be partially overcome through incentives and governance mechanisms [107, 206].

The complex interactions among technology, organization, and the environment in determining adoption highlight the importance of contextually nuanced and integrated conceptual frameworks that consider dependencies across multiple attributes. Future quantitative analyses can explore moderating factors within different environments, while qualitative research adds depth and situational understanding. Policy efforts aim to strike a calibrated balance between bottom-up experimentation and top-down guidance to address constraints. As a result, existing insights provide initial empirical validation of relationships, but also indicate the interactive complexities among various attributes, thereby warranting continued mixed-methods examination at the intersection of emerging technologies and specific contextual factors, such as the digital transformation in developing nations. The present study aims to contribute to this field by analyzing the determinants of blockchain adoption in China's business sustainability accounting environment.

## 5.5. Adoption of cloud computing for sustainability accounting

The implementation of cloud solutions is driven by external factors as well [207]. Nevertheless, Pitt et al. [208] suggested that different types of stakeholders have varying effects on

**Table 4. Analysis of factors influencing cloud computing adoption for sustainability accounting.**

| Study | Sample | Location | Characteristics | Effect Size | Significance |
|---|---|---|---|---|---|
| [142] | 52 companies | India | Resources | $\beta = 0.72$ | $p < 0.001$ |
| [210] | 80 companies | Global | Top management | $\beta = 0.59$ | $p < 0.01$ |
| [211] | 120 industries | Singapore | Risk exposure | $\beta = 0.47$ | $p < 0.05$ |
| [212] | 160 industries | Hong Kong | Industry | $\beta = 0.29$ | non-significant |
| Pooled effect | 139 studies | Global | Various | $\beta = 0.56$ | $p < 0.01$ |

collaborative efforts in alternative energy projects. The significance of these external influences is likely to be influenced by the alignment of goals within complex accountability networks [209].

In a quantitative analysis of existing literature, Table 4 presents an analysis of 139 independent samples that examine factors influencing the adoption of cloud computing for sustainability accounting.

The adoption of cloud solutions is influenced by intrinsic attributes such as financial resources, strategic support, and technical expertise, which have a positive association. Situational factors, such as higher exposure levels leading to material concerns or pressures from stakeholders, can increase the likelihood of adoption. However, other contextual factors require further investigation. The challenges of standardization in dealing with complex systems call for the use of specialized conceptual frameworks.

Qualitative considerations provide insights into the drivers and barriers of adoption through theoretical perspectives. Beaudry and Pinsonneault's coercion-identification model explains how regulatory and normative pressures can either encourage or discourage behaviors in the face of conflicting interests [213]. The institutional logics framework describes how organizations align their preferences with prevailing norms within specific fields [214]. However, there is still a need for a better understanding of the underlying combinations of attributes that drive adoption at a micro level [215].

To gain a deeper understanding of the underlying mechanisms, future research could employ multi-level analyses to operationalize constructs. Integrating ethnographic insights as a cross-validation method could enhance the quantitative explanations. Scholars have advocated for context-sensitive approaches that balance predictive generalizability with explanatory depth through mixed sequential designs [216, 217]. Tailored frameworks tailored to specific contexts may provide more nuanced insights compared to externally developed perspectives.

Therefore, the literature highlights the need to investigate attribute synergies within specific contexts while also recognizing the evolving nature of conceptual lenses that incorporate technological transitions. This study emphasizes the importance of empirical exploration that spans corporate responsibility, emerging solutions, and developing economies undergoing digital transformations. By addressing gaps with contextual sensitivity, theoretical advancements can be made in response to the growing demands for transparency.

## 6. Discussion

This section aims to interpret the main findings of this study related to the factors that influence the adoption of blockchain and cloud-based technologies for sustainability accounting in China. The discussion focuses on analyzing emerging trends, contextual factors, and novel insights generated, while also emphasizing the identified theoretical connections.

In section 5, based on the methodology employed, extensively reviewed the existing literature and generated novel empirical insights through both quantitative and qualitative analyses. The findings consistently indicated a positive correlation between firm size and adoption behaviors. However, it became evident that firm size alone is insufficient to predict outcomes

due to the interdependencies of various attributes, which are influenced by contextual demands. The impact of industry type on adoption behaviors was found to be small yet significant, although inconsistencies highlighted the limitations of relying solely on industry classifications. The relationship between profitability and adoption exhibited nuanced and contingent patterns, necessitating a deeper examination of multifaceted perspectives related to resources and legitimacy, which are moderated by market-specific factors.

By examining the determinants of adoption through theoretical frameworks, explanatory possibilities can be explored, but it is crucial to consider contextual adjustments [189]. The garbage can model emphasizes that decision-making results from the interactions among locally encountered problems, solutions, participants, and opportunities [205]. Institutional logics characterize organizational preferences that align with field-specific norms [214]. Resource-based views focus on how resources guide activities through cost-benefit considerations [197].

Previous empirical studies have indicated that adoption of blockchain and cloud-based technologies for sustainability accounting is influenced by a combination of interactive attributes. The main findings of this study further support the relationships between technical capabilities, financial resources, and the substantiation of reporting (Section 5). However, divergent outcomes observed across different industries and regions have highlighted the complexities arising from boundary conditions, necessitating a deeper conceptualization through region-specific perspectives. The development of contextually-rich, multi-level models that integrate diverse moderating pathways can offer improved predictive and explanatory capabilities to assist stakeholders.

The rapid advancement of technology often surpasses the ability of humans and institutions to adapt [218]. The findings of this study underscore the role of technical uncertainty as a deterrent to the diffusion of these technologies, emphasizing the need for skill development to effectively seize emerging opportunities amidst uncertainties (Section 5.4). Arguments related to legitimacy suggest that addressing stakeholder accountability norms in a responsive manner can facilitate the diffusion process [198]. However, it is important to recognize that stakeholder configurations vary across regions, necessitating the development of region-specific theories to enhance our understanding of the phenomenon [199, 219].

Adapting frameworks to accommodate evolving sustainability challenges and technologies is crucial. Policy interventions can strike a balance between experimentation and guidance while involving diverse stakeholders. Collaborative efforts between academia and businesses can help overcome organizational assimilation and change management hurdles arising from digitization. Future studies should explore variations in technology usage based on industry dynamics, resource availability, and e-readiness levels. Tackling complex adoption contingencies necessitates flexible, multi-disciplinary cooperation.

This study provides initial empirical validation and practical insights into under-explored areas at the intersection of sustainability reporting, emerging technologies, and developmental transitions (Section 8). However, conducting replications in diverse contexts could yield more robust and generalizable implications. Addressing limitations through mixed sequential research designs enhances reliability, while contextualized case comparisons contribute to a deeper interpretation of findings. Ongoing investigations at these intersections hold the potential for further advancements in policy and theory, particularly in response to the increasing demands for transparency.

## 7. Implications and recommendations

This section aims to derive practical implications from the findings and develop strategies, standards, and initiatives to encourage the adoption of sustainable accounting practices

through the utilization of blockchain and cloud technologies. It outlines the necessary actions to be taken by various stakeholders.

The consistent positive relationship between firm size and technology usage suggests the need for customized support to facilitate the digital transition of smaller firms, ensuring the comprehensive advancement of sustainability across diverse industries. Tailored capacity building initiatives can address resource constraints by promoting incremental experimentation rather than abrupt transformations [70, 168, 220].

Considering the limitations of industry classifications in predicting behaviors, a flexible regulatory approach is more suitable to accommodate diverse requirements. Continuous guidance should be provided to incorporate evolving understandings of materiality. Standardization efforts should be approached cautiously to accommodate varying rates of innovation, while also preserving competitive advantages [69, 146, 219].

Mixed findings regarding profitability highlight the need for context-specific support, taking into account industries and national institutional contexts that influence perceived legitimacy and resource constraints, which in turn affect adoption behaviors. Policies should emphasize the importance of legitimate operations aligned with stakeholder expectations, promoting sustained profitability and aligning interests [83, 221–223].

Technical challenges, particularly integration complexities, highlight the importance of collaborative experimentation rather than making premature prediction-driven investments. Skill development through incentivized apprenticeships can partially address the scarcity of expertise. Change stewardship involves gradually integrating emerging solutions [23, 224–226].

Theoretical perspectives suggest that digitization requires organizational culture shifts, including changes in decision-making processes and mental frameworks. Effective leadership prioritizes cooperative decision-making processes that balance bottom-up experimentation with top-down directives to address constraints. Stakeholder networks play a crucial role in diagnosing the interplay of various attributes that differentiate contexts [226–228].

Policy formulation involves striking a balance between flexibility and prescriptiveness. Guidelines are designed to foster fairness and encourage voluntary standards that accommodate diversities, while also signaling expectations for progressive enhancement of quality. Regulations establish minimum requirements that gradually increase over time, allowing for flexible revision cycles as capabilities permit. Independent assurance processes verify compliance while empowering incremental improvements [4, 33, 62].

Incentivized pilot programs facilitate the introduction of new approaches by providing infrastructure and digital literacy investments, reducing barriers, especially for SMEs and startups. Cross-sector learning initiatives expedite the diffusion of skills through cooperative efforts. Investor demands for social responsibility promote the valuation of sustainability practices. Secure and accessible standardization facilitates seamless reporting processes [32, 207, 228].

Hence, achieving technical and organizational transformation requires long-term commitments from multiple stakeholders. Collaborative efforts among stakeholders advance sustainability accounting and disclosure practices, ensuring an equitable transition amidst digital shifts and heightened expectations for transparency on a continually evolving global scale. Fig 4 shows a comprehensive conceptual overview of the significant implications and recommended strategies derived from the research findings on sustainability accounting practices. The visual representation categorizes key facets related to regulatory approaches, capacity building support, collaborative efforts, incentivization programs, and organizational transformation. By mapping the interconnections between these elements, actionable guidance is synthesized for practitioners and policymakers aiming to promote the adoption of emerging

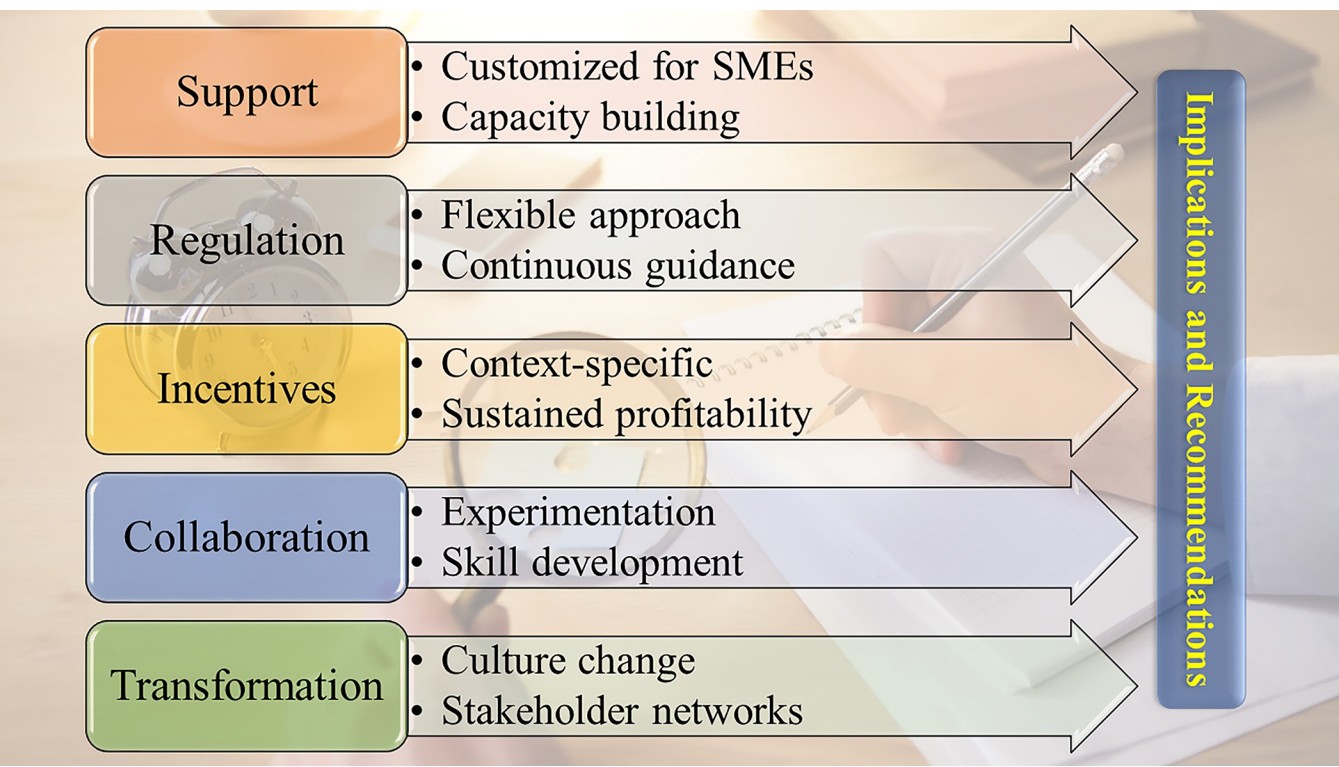

**Fig 4. Framework of implications and recommendations for advancing sustainability accounting practices.**

technologies. The model facilitates the derivation of a multifaceted set of recommendations by considering diverse perspectives encompassing regulation, economics, technology, and organizational behavior. This conceptual perspective informs the formulation of balanced policies and change management strategies to effectively guide the transition towards digitally-enabled sustainability accounting on a global scale.

## 8. Conclusions

The primary objective of this research was to examine the factors that drive and hinder the adoption of emerging blockchain and cloud-based technologies for sustainability accounting in Chinese businesses. This study aimed to contribute empirically and theoretically by addressing existing gaps in knowledge through a thorough mixed-methods investigation within China's evolving context.

Through both quantitative and qualitative analyses, this research generated fresh insights into the relationships among adoption determinants. The findings revealed that firm size, resources, and skills exhibited a positive association with technology usage, although the specific contextual factors necessitated nuanced conceptualizations. Industry classifications also played a role, albeit with small yet significant effects, emphasizing the need to avoid overgeneralization due to heterogeneity. The study further highlighted the intricate and contingent linkages between profitability and technology adoption, which warranted a situational examination.

The adoption of blockchain technology was found to be significantly correlated with capabilities, resources, skills, and regulatory factors. However, the complexities of the context, as evidenced by heterogeneity, indicated interdependencies among these attributes. On the other

hand, cloud computing adoption showed a positive association with resources, management support, and risk exposures, although the impact of industry factors remained inconclusive and necessitated qualitative investigation.

From a theoretical perspective, the study identified legitimacy and resources as key starting points for understanding the adoption process, but emphasized the importance of contextualization. The decision-making processes were found to be influenced by institutional logics and the lens of the garbage can model, which highlighted the role of localized interactions. The study also shed light on the barriers to technological assimilation, such as uncertainties related to disruption and challenges in integration.

The findings of this study not only provided empirical validation of relationships but also underscored the need for a comprehensive understanding of attribute synergies within the evolutionary contexts of these technologies. Situational variations highlighted the importance of considering contingencies and incorporating moderators through microfoundational and multi-level analytical approaches. Qualitative probing complemented the interpretations of the findings.

In practical terms, customized capacity building was identified as an effective strategy for addressing resource constraints through phased experimentation. Flexible and industry-specific guidance was recommended to promote continuous progress that incorporates evolving understandings. Collaborative piloting was seen as a way to introduce new approaches and invest in literacy, particularly benefiting underrepresented stakeholders.

Moving forward, mixed sequential designs were suggested to reconcile predictive generalizability with explanatory depth. Regionalized case comparisons were proposed to explore boundary conditions, while cooperative policy formulation was deemed necessary to strike a balance between participation and standardization. Conceptual lenses that incorporate technological transitions were seen as crucial for understanding and keeping up with accelerating expectations. Addressing the limitations of this study promises further theoretical contributions that optimize sustainability accounting through responsible innovation on a global scale.

In conclusion, this research bridged knowledge gaps through a comprehensive investigation focused on the adoption attributes of blockchain and cloud computing in the context of sustainability accounting in China. The findings of this study make novel empirical and theoretical contributions, emphasizing the interactive complexities of these attributes that require nuanced and multi-level conceptualizations tailored to local contexts undergoing digital transformation. By addressing contextual sensitivities, this research aims to contribute to the ongoing advancement in meeting the growing demands for transparency through interdisciplinary cooperation.

## Supporting information

**S1 File. Table of references information.** URL, DOI, and reference information for Tables 1–4 in the study.
(XLSX)

## Author Contributions

**Conceptualization:** Zhouyu Tian, Lening Qiu, Litao Wang.

**Formal analysis:** Zhouyu Tian, Lening Qiu.

**Investigation:** Zhouyu Tian, Lening Qiu, Litao Wang.

**Methodology:** Zhouyu Tian, Litao Wang.

**Validation:** Zhouyu Tian, Lening Qiu, Litao Wang.

**Visualization:** Zhouyu Tian, Lening Qiu.

**Writing – original draft:** Litao Wang.

**Writing – review & editing:** Zhouyu Tian, Lening Qiu.

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
