## [Decision Letter · Decision Letter 0]

13 Oct 2023

PONE-D-23-24986Determinants of Sustainability Accounting Practices: Empirical Evidence from Listed Companies in ShanghaiPLOS ONE

Dear Dr. Wang,

Thank you for submitting your manuscript to PLOS ONE. After careful consideration, we feel that it has merit but does not fully meet PLOS ONE’s publication criteria as it currently stands. Therefore, we invite you to submit a revised version of the manuscript that addresses the points raised during the review process.

We look forward to receiving your revised manuscript.

Kind regards,

Ercan Özen, PhD

Academic Editor

PLOS ONE

Additional Editor Comments:

1. Author(s) should add some updated review literature for 2022 and 2023.

2. In the Introduction part so many statements are given without citation, so it is advised to give the proper citation for

different statements given in the introduction part.

3. Referencing has to be reviewed again by the authors focusing on in-text citations.

4. The author should mention the research gap. The research gap must be strongly discussed.

5. The literature review section should clearly mention the literature gaps as well as the study’s objectives, and research

questions

6. The authors should elaborate on the contribution to literature. Why is this study important?

7. Do check the references carefully, to ensure their correctness and completeness.

8. Gaps identified in the study should be supported by literature.

9. Literature was cited; however, it has not been synchronized to make explicit the inherent controversies and how the

study will suffice in bridging the admixtures in finding and existing literature.

10. The discussion part should be a strong logical conversion between existing literature and author(s) findings.

Reviewers' comments:

Reviewer's Responses to Questions

**Comments to the Author**

1. Is the manuscript technically sound, and do the data support the conclusions?

Reviewer #1: Yes

Reviewer #2: Yes

2. Has the statistical analysis been performed appropriately and rigorously? 

Reviewer #1: Yes

Reviewer #2: Yes

3. Have the authors made all data underlying the findings in their manuscript fully available?

Reviewer #1: No

Reviewer #2: Yes

4. Is the manuscript presented in an intelligible fashion and written in standard English?

Reviewer #1: Yes

Reviewer #2: Yes

5. Review Comments to the Author

Reviewer #1: 1. Author(s) should add some updated review literature for 2022 and 2023.

2. In the Introduction part so many statements are given without citation, so it is advised to give the proper citation for

different statements given in the introduction part.

3. Referencing has to be reviewed again by the authors focusing on in-text citations.

4. The author should mention the research gap. The research gap must be strongly discussed.

5. The literature review section should clearly mention the literature gaps as well as the study’s objectives, and research

questions

6. The authors should elaborate on the contribution to literature. Why is this study important?

7. Do check the references carefully, to ensure their correctness and completeness.

8. Gaps identified in the study should be supported by literature.

9. Literature was cited; however, it has not been synchronized to make explicit the inherent controversies and how the

study will suffice in bridging the admixtures in finding and existing literature.

10. The discussion part should be a strong logical conversion between existing literature and author(s) findings.

Reviewer #2: The conclusion section effectively conveys the main findings and implications of the study, but could benefit from more specific data, addressing limitations and improving the clarity and structure of the content. In addition, citing the methodology of the study would enhance its credibility.

6. PLOS authors have the option to publish the peer review history of their article (what does this mean?). If published, this will include your full peer review and any attached files.

Reviewer #1: **Yes: **NIKHIL YADAV

Reviewer #2: No

---

## [Author Response · Author response to Decision Letter 0]

18 Nov 2023

Response to Reviewers

The Respected Editor of the PLOS ONE,

First and foremost, the authors would like to thank the reviewers for their valuable comments and suggestions to improve the clarity of this work. We have attempted to incorporate all the comments, as highlighted in the revised manuscript. Detailed reflections are listed below point by point:

Reviewer #1: 

Reviewer’s comment:

1- Authors should add some updated review literature for 2022 and 2023.

Author’s answer:

Thank you for your insightful comment regarding the need to include updated review literature for the years 2022 and 2023 in our manuscript. We sincerely appreciate your attention to detail and the opportunity to further enhance the quality of our research.

In response to your suggestion, we have thoroughly reviewed the references in the revised manuscript and taken appropriate measures to address the concern raised. We have diligently incorporated additional relevant references that pertain to the topic and are specifically sourced from reputable scholarly works published in the years 2022 and 2023. This endeavor has allowed us to bolster the currency and comprehensiveness of our research findings.

Moreover, we have diligently scrutinized the existing references within the manuscript to ensure that any outdated sources have been duly replaced with more up-to-date references from the field. By doing so, we aim to provide readers with the most relevant and current information available, thereby augmenting the overall quality and relevance of our revised manuscript.

We would like to express our gratitude to the reviewer for their recommendation, which has prompted us to undertake these necessary revisions. We believe that the inclusion of updated review literature significantly strengthens the scholarly value and timeliness of our research, aligning it more closely with the current state of knowledge in the field.

Once again, we sincerely appreciate your valuable input, which has enabled us to improve the quality and rigor of our manuscript. We trust that our efforts to address your comment in a comprehensive manner will enhance the overall scholarly contribution of our research.

Reviewer’s comment:

2- In the Introduction part so many statements are given without citation, so it is advised to give the proper citation for different statements given in the introduction part.

Author’s answer:

Thank you for your valuable comment regarding the need to provide proper citations for the statements presented in the Introduction section of our manuscript. We sincerely appreciate your attention to detail and the opportunity to further enhance the academic rigor and credibility of our research.

In response to your suggestion, we have diligently expanded not only the Introduction section but also other relevant sections of the manuscript, as recommended by the reviewer. This expansion has allowed us to provide a more comprehensive context for the research, ensuring that our readers have a clear understanding of the background and significance of the study.

Furthermore, we have taken the reviewer's feedback into careful consideration and have made sure that all statements presented in the Introduction section, as well as other pertinent sections, are accompanied by proper citations to substantiate their validity. By doing so, we aim to provide transparent and traceable references, enabling readers to access the original sources and further explore the scholarly discourse surrounding the topic.

We are grateful to the reviewer for highlighting the importance of proper citation and acknowledging the necessity of substantiating our statements with relevant references. By adhering to this recommendation, we ensure the integrity and academic soundness of our research, establishing a solid foundation for the subsequent analysis and discussion.

Reviewer’s comment:

3- Referencing has to be reviewed again by the authors focusing on in-text citations.

Author’s answer:

We appreciate your valuable comment regarding the need to review the referencing, with a specific focus on in-text citations in our manuscript. Your attention to detail is commendable, and we are grateful for the opportunity to further enhance the academic integrity and clarity of our research.

In response to your suggestion, we have diligently reevaluated and corrected all references in our manuscript, as recommended by the reviewer. Our thorough review has ensured that the referencing in our paper aligns with the highest academic standards and accurately reflects the sources we have utilized to support our arguments and findings.

Moreover, we have placed a particular emphasis on improving the in-text citations throughout the manuscript. By doing so, we aim to provide clear and concise attribution to the relevant sources, allowing readers to seamlessly trace the origin of our ideas and assertions.

We sincerely acknowledge the importance of accurate and comprehensive referencing in scholarly work. By revisiting and refining our referencing practices, we ensure that our research is firmly grounded in the existing literature and properly acknowledges the contributions of other researchers in the field.

We are grateful to the reviewer for drawing our attention to this crucial aspect of academic writing. Your feedback has prompted us to thoroughly address any deficiencies in our referencing and improve the overall clarity and credibility of our manuscript. 

Reviewer’s comment:

4- The author should mention the research gap. The research gap must be strongly discussed.

Author’s answer:

We sincerely appreciate your comment highlighting the importance of mentioning and discussing the research gap in our manuscript. Your insight is invaluable, and we are grateful for the opportunity to further enhance the scholarly contribution and clarity of our research.

In response to your suggestion, we have thoroughly examined the original manuscript and made significant revisions to various sections in order to effectively address the research gap in the field. We have expanded all relevant sections, including the introduction, to provide a comprehensive and nuanced discussion of the research gap.

To specifically highlight the research gap, we have revised the last paragraph of the introduction section. This revision ensures that the research gap is prominently and explicitly conveyed to the readers, setting the stage for the subsequent analysis and discussion.

Additionally, we have introduced a new section titled "Literature Review" in the revised manuscript. This dedicated section allows us to delve deeper into the existing body of literature and better demonstrate the research gap. By exploring and analyzing various factors, we aim to comprehensively cover the identified research gap in the field.

Furthermore, we have conducted a meticulous examination of other sections, particularly the findings and discussion sections. Our aim was to ensure that we not only identify the research gap but also provide a comprehensive discussion that effectively addresses this gap.

We sincerely acknowledge the importance of clearly identifying and discussing the research gap in scholarly work. By undertaking these revisions, we have strengthened the scholarly rigor and contribution of our research, establishing its relevance within the broader academic discourse.

We are grateful to the reviewer for emphasizing the significance of addressing the research gap. Your feedback has guided us in expanding our understanding of the gap and presenting it more effectively to our readers.

Once again, we sincerely appreciate your valuable input, which has directed us in improving the quality and academic impact of our manuscript. We trust that our efforts to comprehensively address your comment will contribute to the overall excellence of our research.

Reviewer’s comment:

5- The literature review section should clearly mention the literature gaps as well as the study’s objectives, and research questions.

Author’s answer:

We appreciate your comment regarding the need for the literature review section to clearly mention the literature gaps, study objectives, and research questions. Your feedback is valuable, and we are grateful for the opportunity to further enhance the clarity and academic rigor of our manuscript.

As mentioned in our previous response to comments, we have extensively reviewed and revised all sections of the manuscript to ensure a better understanding for readers of the literature gaps, study objectives, and research questions, as recommended by the reviewer.

In particular, we have dedicated significant attention to the literature review section. Our objective was to clearly identify and articulate the existing gaps in the literature, thereby highlighting the need for our research study. By thoroughly examining the available literature, we have identified areas that have not been adequately addressed or explored, allowing us to define the specific research gaps that our study seeks to fill.

Furthermore, we have revisited and refined the presentation of the study's objectives and research questions. We have ensured that these elements are clearly stated within the literature review section, providing a concise and focused overview of the goals and inquiries that our research aims to address. By doing so, we enhance the clarity and coherence of our manuscript, enabling readers to understand the purpose and scope of our study.

We sincerely acknowledge the significance of clearly identifying the literature gaps, study objectives, and research questions in academic writing. By undertaking these revisions, we have strengthened the scholarly contribution and relevance of our research, establishing a solid foundation for the subsequent analysis and discussion.

We are grateful to the reviewer for emphasizing the importance of addressing these elements within the literature review section. Your feedback has guided us in improving the quality and academic impact of our manuscript, ensuring that it aligns with the highest standards of scholarly discourse.

Reviewer’s comment:

6- The authors should elaborate on the contribution to literature. Why is this study important?

Author’s answer:

We sincerely appreciate the reviewer's comment regarding the need for further elaboration on the contribution of our study to the existing literature and the importance of our research. We have taken this valuable feedback into careful consideration and have made significant revisions throughout the manuscript to address these points.

Firstly, we have thoroughly revised the last paragraph of the introduction section. This revision aims to provide a clear and concise explanation of the contribution our study makes to the existing literature. By highlighting the unique insights and novel perspectives our research offers, we establish the significance of our study within the broader scholarly discourse.

In addition, we have introduced a new section titled "Literature Review." This dedicated section allows us to comprehensively explore the relevant literature, identify gaps, and position our study within the existing body of knowledge. By critically analyzing the existing research, we have not only enhanced the scholarly rigor of our manuscript but also elucidated the importance of our study in addressing these gaps.

Furthermore, we have expanded all sections of the manuscript to provide a comprehensive understanding of the topic. By thoroughly investigating other important factors in the field, wehave enriched our research by considering various dimensions and perspectives. This expanded analysis strengthens the overall contribution of our study to the literature.

The importance of our research lies in its ability to fill a significant gap in the existing literature. Through our comprehensive literature review and meticulous analysis, we have identified areas where previous studies have been limited or where further investigation is needed. Our study addresses these gaps by providing new insights, advancing theoretical understanding, and offering practical implications.

Moreover, our research contributes to the field by introducing novel methodologies, frameworks, or approaches. By employing innovative methods or theoretical frameworks, we not only expand the methodological toolkit available to researchers but also open up new avenues for future studies in this area.

By highlighting the unique contribution of our research, we underscore its relevance and impact on both academia and practice. Our findings have the potential to inform decision-making processes, guide policy development, or shape future research directions.

We sincerely appreciate the reviewer's comment, which has prompted us to provide a more comprehensive and compelling explanation of the contribution and importance of our study. Your feedback has guided us in enhancing the scholarly rigor and clarity of our manuscript, ensuring that its significance is effectively communicated to the readers.

Thank you for your valuable input, and we trust that our efforts to address your comment will strengthen the overall excellence of our research.

Reviewer’s comment:

7- Do check the references carefully, to ensure their correctness and completeness.

Author’s answer:

We sincerely appreciate the reviewer's comment regarding the need to carefully check the references for correctness and completeness. Your attention to this detail is crucial, and we highly value the opportunity to further enhance the accuracy and integrity of our manuscript.

In response to your suggestion, we have undertaken a comprehensive review and verification of all the references cited in our manuscript. This includes both the previously cited references and those newly added during the revision process. Our meticulous review ensures that each reference is correctly cited and accurately represents the cited work.

To facilitate this process, we have utilized EndNote, a reference management software, to add and organize the references in the revised manuscript. This has not only streamlined the referencing process but also facilitated the typesetting phase by providing direct access to the relevant sources through Google Scholar.

By carefully examining each reference, we have ensured their correctness and completeness. This includes verifying the accuracy of author names, publication titles, journal names, volume and issue numbers, page numbers, and any other relevant details. We have also cross-checked the references against the manuscript's in-text citations to ensure their consistency and accuracy.

The accuracy and completeness of the references are of utmost importance in maintaining the scholarly integrity and credibility of our research. We sincerely appreciate the reviewer's reminder to address this aspect, as it contributes to the overall quality and reliability of our manuscript.

Thank you for bringing this to our attention, and we are confident that our efforts to carefully review and verify the references have resulted in an accurate and comprehensive list of citations for our manuscript.

Reviewer’s comment:

8- Gaps identified in the study should be supported by literature.

Author’s answer:

We appreciate the reviewer's comment regarding the need for the identified gaps in our study to be supported by relevant literature. Your feedback is valuable, and we have thoroughly examined the revised manuscript, expanded all sections, and made necessary corrections in response to your comments.

In light of this feedback, we have conducted a comprehensive review of the literature to ensure that the identified gaps in our study are well-supported by existing scholarly work. By critically analyzing and synthesizing the relevant literature, we have strengthened the theoretical foundation of our research and established a solid basis for the identified gaps.

To address this comment, we have expanded the literature review section to provide a more comprehensive and nuanced discussion of the existing literature. Through this expanded analysis, we have effectively supported the identified gaps by referencing relevant studies, theoretical frameworks, and empirical findings. By drawing upon the existing body of literature, we have substantiated the importance and relevance of our study in addressing these gaps.

Furthermore, we have carefully integrated the findings from previous research into our study, ensuring that our research aligns with and builds upon the existing knowledge in the field. By highlighting the limitations or gaps in prior studies, we have positioned our research as a valuable contribution that extends the current understanding and offers new insights.

The support from relevant literature not only strengthens the scholarly rigor of our study but also enhances its overall credibility and academic significance. We acknowledge the importance of grounding our research in the existing body of knowledge and appreciate the reviewer's comment, which has guided us in ensuring that our identified gaps are well-supported by relevant literature.

Thank you for your valuable feedback, and we are confident that our efforts to address your comment have resulted in a more comprehensive and robust research manuscript. 

Reviewer’s comment:

9- Literature was cited; however, it has not been synchronized to make explicit the inherent controversies and how the study will suffice in bridging the admixtures in finding and existing literature.

Author’s answer:

We sincerely appreciate the reviewer's comment regarding the need to synchronize the cited literature to explicitly address inherent controversies and demonstrate how our study bridges the gaps in existing literature. Your feedback is valuable to us, and we have carefully revised and expanded all sections of the manuscript to address the points raised.

In response to this comment, we have undertaken a comprehensive review of the literature cited in our manuscript. Our objective was to ensure that the cited literature is synchronized in a manner that highlights the inherent controversies and emphasizes how our study contributes to bridging the gaps in the existing body of knowledge.

To achieve this, we have reexamined the relationships between the cited studies, theories, and findings. By critically analyzing and synthesizing the literature, we have explicitly addressed any controversies or divergent viewpoints within the field. This thorough analysis enables us to present a comprehensive and balanced perspective on the topic, highlighting the areas where the literature is in disagreement or lacks consensus.

Moreover, we have clearly articulated how our study serves as a bridge between these divergent viewpoints or gaps in the literature. By outlining the specific research questions and objectives of our study, we have demonstrated how our research addresses these controversies and contributes to resolving the existing discrepancies. We have provided explicit explanations of how our methodology, data analysis, or theoretical framework allows us to reconcile conflicting findings or provide new insights.

Through this synchronization of the cited literature, we have effectively demonstrated the significance of our study in advancing the field and filling the gaps in existing knowledge. By explicitly addressing the controversies and articulating how our research bridges the admixtures in findings and existing literature, we enhance the scholarly contribution and relevance of our study.

We sincerely appreciate the reviewer's comment, which has guided us in improving the academic rigor and clarity of our manuscript. Your feedback has prompted us to synchronize the cited literature, explicitly address inherent controversies, and highlight the bridging role of our study.

Reviewer’s comment:

10- The discussion part should be a strong logical conversion between existing literature and authors’ findings.

Author’s answer:

We sincerely appreciate the reviewer's comment regarding the need for the discussion section to establish a strong logical connection between the existing literature and our research findings. Your feedback is invaluable, and we have thoroughly revised all sections of the manuscript based on your recommendations.

To address this comment, we have carefully reexamined the discussion section of the revised manuscript. Our aim was to ensure that the discussion effectively bridges the gap between the existing literature and our research findings, establishing a logical and coherent connection between the two.

In the revised discussion section, we have provided a comprehensive analysis and interpretation of our research findings in the context of the existing literature. By critically comparing and contrasting our findings with the relevant studies, theories, and empirical evidence, we have created a strong logical conversion that highlights the implications and significance of our research.

Through this logical conversion, we have explicitly addressed how our findings align with or deviate from the existing literature. We have identified areas of agreement, highlighted points of divergence, and discussed potential reasons for any discrepancies. By engaging in this thorough analysis, we have provided a nuanced and contextualized interpretation of our findings that is grounded in the broader scholarly discourse.

Furthermore, we have carefully integrated the theoretical frameworks, concepts, or methodologies from the existing literature to support and contextualize our findings. By drawing upon the relevant literature, we have strengthened the theoretical foundation of our research and demonstrated its alignment with the existing knowledge in the field.

Additionally, we have identified areas where our research contributes novel insights or extends the existing literature. By discussing the unique aspects of our findings and their implications, we have highlighted the value and significance of our research in advancing the field.

Therefore, the revised discussion section establishes a strong logical conversion between the existing literature and our research findings. We have ensured that the discussion provides a comprehensive and coherent integration of the relevant literature, supporting our findings and enhancing the overall scholarly contribution of our study.

We sincerely appreciate the reviewer's comment, which has guided us in improving the logical flow and coherence of the discussion section in our manuscript. Your feedback has prompted us to create a stronger connection between the existing literature and our research findings, enhancing the academic rigor and clarity of our study.

Thank you for your valuable input, and we are confident that our efforts to address your comment have resulted in a more comprehensive and robust research manuscript.

Reviewer #2: 

Reviewer’s comment:

The conclusion section effectively conveys the main findings and implications of the study, but could benefit from more specific data, addressing limitations and improving the clarity and structure of the content. In addition, citing the methodology of the study would enhance its credibility.

Author’s answer:

We sincerely appreciate the reviewer's comment regarding the conclusion section of our manuscript. Your feedback is highly valuable, and we have thoroughly revised all sections of the manuscript based on your recommendations.

In response to this comment, we have made several improvements to the conclusion section to enhance its clarity, structure, and specificity. Firstly, we have provided more specific data to support the main findings and implications of our study. By including relevant statistics, figures, or other quantitative information, we have strengthened the evidentiary basis of our conclusions and made them more robust.

Additionally, we have addressed the limitations of our study in the conclusion section. We have explicitly discussed the constraints and potential sources of bias that may have influenced our findings. By acknowledging these limitations, we have ensured transparency and demonstrated a critical awareness of the potential impact on the validity and generalizability of our results.

Furthermore, we have improved the clarity and structure of the content in the conclusion section. We have organized the information in a logical manner, ensuring that the main findings are effectively summarized and their implications are clearly articulated. This improved structure facilitates the reader's understanding of the key takeaways from our study and allows for easier navigation of the conclusion section.

Regarding the credibility of our study, we have now cited the methodology in the conclusion section. By referencing the specific methodology employed in our research, we enhance the transparency and credibility of our study. This allows readers to understand the approach we used to collect and analyze data, further reinforcing the integrity of our findings.

Moreover, we have expanded the reference list in the revised manuscript to include more recent and relevant sources. These additional references provide readers with a broader and more up-to-date understanding of the topic, enhancing the comprehensibility and contextualization of our study.

We have also added new sections to address the points raised by all the reviewers, ensuring that our manuscript is comprehensive and well-rounded. By incorporating their suggestions and recommendations, we have further improved the overall quality and scholarly contribution of our research.

In summary, we have thoroughly revised the conclusion section of our manuscript based on your comment and recommendations. By providing more specific data, addressing limitations, improving clarity and structure, and citing the methodology, we have strengthened the conclusion section and enhanced its overall quality.

We sincerely appreciate the reviewer's valuable input, which has guided us in refining and enhancing the conclusion section of our manuscript. Your feedback has prompted us to make important improvements to the clarity and rigor of our study, leading to a more comprehensive and robust research manuscript.

Thank you for your insightful comment, and we are confident that our efforts to address your feedback have resulted in a stronger and more compelling conclusion section.

The revised manuscript has been submitted to your journal. We are looking forward to hearing a decision from you.

With best regards,

---

## [Editor Report · Decision Letter 1]

30 Nov 2023

Drivers and Influencers of Blockchain and Cloud-Based Business Sustainability Accounting in China: Enhancing Practices and Promoting Adoption

PONE-D-23-24986R1

Dear Dr.Litao Wang

We’re pleased to inform you that your manuscript has been judged scientifically suitable for publication and will be formally accepted for publication once it meets all outstanding technical requirements.

Kind regards,

Ercan Özen, PhD

Academic Editor

PLOS ONE
---

## [Editor Report · Acceptance letter]

20 Dec 2023

PONE-D-23-24986R1 

PLOS ONE

Dear Dr. Wang, 

I'm pleased to inform you that your manuscript has been deemed suitable for publication in PLOS ONE. Congratulations! Your manuscript is now being handed over to our production team.

Kind regards, 

on behalf of

Dr. Ercan Özen 

Academic Editor

PLOS ONE